# Brachytherapy Approach Using ^177^Lu Conjugated Gold Nanostars and Evaluation of Biodistribution, Tumor Retention, Dosimetry and Therapeutic Efficacy in Head and Neck Tumor Model

**DOI:** 10.3390/pharmaceutics13111903

**Published:** 2021-11-09

**Authors:** Min-Ying Lin, Hsin-Hua Hsieh, Jyh-Cheng Chen, Chuan-Lin Chen, Nin-Chu Sheu, Wen-Sheng Huang, Shinn-Ying Ho, Ting-Wen Chen, Yi-Jang Lee, Chun-Yi Wu

**Affiliations:** 1Department of Biomedical Imaging and Radiological Sciences, National Yang Ming Chiao Tung University, Taipei Branch, Taipei 112, Taiwan; milo841120.be09@nycu.edu.tw (M.-Y.L.); alexa.be09@nycu.edu.tw (H.-H.H.); jcchen@ym.edu.tw (J.-C.C.); clchen2@ym.edu.tw (C.-L.C.); mir870106.y@nycu.edu.tw (N.-C.S.); 2Department of Nuclear Medicine, Taipei Medical University Hospital, Taipei 11031, Taiwan; wshuang01@gmail.com; 3Institute of Bioinformatics and Systems Biology, National Yang Ming Chiao Tung University, Hsinchu Branch, Hsinchu 30068, Taiwan; syho@nctu.edu.tw (S.-Y.H.); dodochen@nctu.edu.tw (T.-W.C.); 4Department of Biological Science and Technology, National Yang Ming Chiao Tung University, Hsinchu Branch, Hsinchu 30068, Taiwan; 5Cancer Progression Research Center, National Yang Ming Chiao Tung University, Taipei Branch, Taipei 112, Taiwan

**Keywords:** brachytherapy, ^177^Lu-DTPA-pAuNS, head and neck cancer, photothermal therapy

## Abstract

Brachytherapy can provide sufficient doses to head and neck squamous cell carcinoma (HNSCC) with minimal damage to nearby normal tissues. In this study, the β^−^-emitter ^177^Lu was conjugated to DTPA-polyethylene glycol (PEG) decorated gold nanostars (^177^Lu-DTPA-pAuNS) used in surface-enhanced Raman scattering and photothermal therapy (PTT). The accumulation and therapeutic efficacy of ^177^Lu-DTPA-pAuNS were compared with those of ^177^Lu-DTPA on an orthotopic HNSCC tumor model. The SPECT/CT imaging and biodistribution studies showed that ^177^Lu-DTPA-pAuNS can be accumulated in the tumor up to 15 days, but ^177^Lu-DTPA could not be detected at 24 h after injection. The tumor viability and growth were suppressed by injected ^177^Lu-DTPA-pAuNS but not nonconjugated ^177^Lu-DTPA, as evaluated by bioluminescent imaging. The radiation-absorbed dose of the normal organ was the highest in the liver (0.33 mSv/MBq) estimated in a 73 kg adult, but that of tumorsphere (0.5 g) was 3.55 mGy/MBq, while intravenous injection of ^177^Lu-DTPA-pAuNS resulted in 1.97 mSv/MBq and 0.13 mGy/MBq for liver and tumorsphere, respectively. We also observed further enhancement of tumor-suppressive effects by a combination of ^177^Lu-DTPA-pAuNS and PTT compared to ^177^Lu-DTPA-pAuNS alone. In conclusion, ^1^^77^Lu-DTPA-pAuNS may be considered as a potential radiopharmaceutical agent for HNSCC brachytherapy.

## 1. Introduction

Head and neck squamous cell carcinoma (HNSCC) ranks sixth in incidence around the world. The 5-year survival rate of HNSCC is around 65% and locoregional recurrence is believed to be a major limitation of various treatments, including surgical and nonsurgical strategies [1,2]. Unlike other cancers that occur in parenchymal organs, the dissemination of HNSCC is mainly caused by local invasion instead of distant metastasis [3]. Local invasion of HNSCC is also a risk factor of recurrence after therapy, and the mechanism of local invasion has been reported to be associated with epithelial-mesenchymal transition (EMT) of tumor cells [4,5]. A recent report demonstrated that EMT can be regarded as a prognostic factor for HNSCC patients who undergo chemoradiotherapy [6]. Rhenium-188 composed of radiopharmaceuticals with diagnostic and therapeutic capacity also exhibit significant effects on repressing EMT markers of orthotopic HNSCC tumor model [7]. Radiopharmaceuticals or radionuclides are considered as neoadjuvant high-dose-rate brachytherapy for breast cancer or cervical cancer [8,9]. For recurrent HNSCC, high-dose-rate brachytherapy has also been reported to improve overall survival without unacceptable acute toxicity [10,11]. Therefore, the development of specific radiopharmaceuticals is needed for local treatment of advanced HNSCC and to reduce potential side effects raised by radioactivity.

Certain radionuclides emit both moderate- or high-energy particles and low-energy γ-rays that can be used for in vivo diagnosis and therapy concomitantly, the so-called theranostics. Lutetium-177 (^177^Lu) is probably the most attractive theranostic radionuclide contemporarily as it shows therapeutic advantages in various fields of preclinical and clinical communities [12,13,14]. ^177^Lu exists in an oxidative state of +3 charge that allows it to be easily radiolabeled to a variety of molecular carriers, and this property is important for the design of targeted radionuclide therapy (TRT) [15]. ^177^Lu emitted 78% of 0.497 MeV β^−^ particles (E_β(max)_) and 11% of 0.208 MeV γ-rays that can be used for theranostics and dosimetry [16]. The therapeutic efficacy of moderate-energy β^−^ particles can be compensated by a relatively long half-life (6.65 days) and high activity of ^177^Lu during the preparation of radiopharmaceuticals and administration to patients. The mean emission range of β^−^ particles delivered by ^177^Lu is 670 μm [15]. The diameter of one mammalian cell is around 100 μm, therefore, ^177^Lu is ideal for the treatment of local invaded tumor cells and surface tumors with low damage to nearby normal tissues. Therefore, the physical and chemical characteristics of ^177^Lu seem applicable for its administration in local advanced HNSCC. However, little is known if ^177^Lu formed radiopharmaceuticals can be used for local theranostics of HNSCC, at least in a preclinical model.

Gold nanostars (AuNS) belong to a specific shape of gold nanoparticles (GNP) that constitute a solid metal core decorated with multiple sharp branches [17]. AuNS have been used as a substrate for surface-enhanced Raman scattering (SERS), localized surface plasmon resonance (LSPR) sensing, and refractive index (RI) sensing [18,19,20]. Studies have also demonstrated that AuNS are biocompatible and their biomedical applications, including diagnosis, bioimaging, photothermal therapy, and target drug delivery are of interest [21,22,23,24]. For instance, AuNS can be used as photothermal agents for cancer treatment because of the high conversion efficiency raised by the near-infrared (NIR) laser [25,26]. A similar approach using NIR light irradiation can also stimulate the release of a chemotherapeutic agent coated on AuNS [27]. However, whether AuNS can be used as a nanocarrier for radionuclides and applied in tumor imaging and therapy remains to be addressed. ^177^Lu-labeled GNP is found to be highly effective for inhibiting the growth of breast cancer in tumor-bearing mice by intratumoral administration [28], the radiolabeling of AuNS is also speculated to be also applicable to theranostics in the HNSCC tumor model.

Based on the promising theranostic properties of ^177^Lu and the nature of the gold nanostars, we aim to develop PEGylated coated AuNS for conjugating ^177^Lu via a DTPA chelator and determine their biodistribution by using noninvasive imaging as well as the therapeutic efficacy. To the best of our knowledge, this is the first report proposing the preclinical application of ^177^Lu labeled AuNS in the human HNSCC tumor model.

## 2. Materials and Methods

### 2.1. Preparation and Characterization of ^177^Lu-Labeled pAuNS (^177^Lu-DTPA-pAuNS)

The manufacturing procedure of gold nanostars (AuNS) has been described previously [26]. Poly(ethylene glycol) methyl ether thiol (45 mg, SH-PEG_6000_-CH_3_, Sigma-Aldrich, St. Louis, MO, USA) and Thiol PEG amine (15 mg, SH-PEG2K-NH_2_, Sigma-Aldrich, St. Louis, MO, USA) were added to the AuNS solution. The mixture was kept at room temperature for 2 h with gentle shaking. After the reaction, the solution was centrifuged at 12,000× *g* for 15 min to remove unconjugated compounds in the supernatant. The pellet was resuspended in ddH_2_O. To this solution, the chelate, p-SCN-Bn-DTPA (0.5 mg, Macrocyclics, Plano, TX, USA) dissolved in 0.05 M of sodium bicarbonate buffer (pH = 9.2), was added and the mixture was reacted at 60 °C for 4 h. The unreacted chelate was removed by centrifugation (12,000× *g*, 15 min) to give DTPA-pAuNS as a pellet, which was washed with ddH_2_O twice for radiolabeling. ^177^Lu-LuCl_3_ (ITG Isotope Technologies Garching GmbH, Garching, Germany) and DTPA-pAuNS (^177^Lu-LuCl_3_/DTPA-pAuNS = 10 μCi/1 μg) were added to a vial containing HEPES buffer (0.1 M, pH = 4.5) and the mixture was allowed to react at 30 °C for 30 min. After the reaction, excess DTPA solution (10 mM) was loaded for a 30-min challenge. Finally, the mixture was centrifuged at 10,000× *g* for 10 min to obtain the product, ^177^Lu-DTPA-AuNS. The labeling efficiency and radiochemical purity were determined by radio-thin layer chromatography (radio TLC) using sodium citrate buffer (0.5 M, pH = 5.0) as mobile phase on an ITLC plate (ITLC-SG, Merck, Darmstadt, Germany).

### 2.2. Cell Lines

Human tongue squamous cell carcinoma SAS cells were obtained from Prof. Muh-Hwa Yang (National Yang Ming Chiao Tung University, Taipei, Taiwan) and SAS-3R cells (a stable cell line with multiple reporter genes, including green fluorescent protein gene, firefly luciferase gene and herpes simplex virus type 1-thymidine kinase gene) harboring a pLT-3R construct with multiple reporter genes were maintained as described in a previous report [29]. Cells were maintained in DMEM (Life Technologies Inc., Carlsbad, CA, USA). All cell lines were supplemented with 10% fetal bovine serum (Thermo Fisher Scientific Inc., Waltham, MA, USA), 2 mM L-glutamine, and 50 U/mL penicillin. Cells were incubated at 37 °C in a humidified incubator containing 5% CO_2_.

### 2.3. Human HNSCC Tumor-Bearing Animal Model

The SAS-3R cells were implanted on BALB/c nude mice with orthotopic injection. The SAS-3R cells were carefully cultured and resuspended in OPTI-MEM (Sigma-Aldrich, St. Louis, MO, USA) at a concentration of 10^6^ cells/50 μL. The mice were anesthetized and placed supine. The mouth was opened and the 27G needle syringe (Terumo Co., Tokyo, Japan) was then inserted for 5 mm and injected through the outer muscle of the lower jaw on the right side of mice. All the experiment animals were approved by the Institutional Animal Care and Use Committee (IACUC No. 1081009).

### 2.4. Analysis of Biodistribution of ^177^Lu-DTPA and ^177^Lu-DTPA-pAuNS

The tumor-bearing mice were randomly assigned to two groups for intratumoral injection of ^177^Lu-DTPA and ^177^Lu-DTPA-pAuNS followed by biodistribution. Mice were sacrificed by CO_2_ asphyxiation to collect organs and tumors after 4, 24, and 72 h of injection. Collected organs were weighed and counted by the γ-scintillation counter (1470 WIZARD Gamma Counter, Wallac, Finland). The results are presented as the percentage injected dose per gram tissue (% ID/g).

### 2.5. MicroSPECT Imaging of ^177^Lu-DTPA-pAuNS in Tumor-Bearing Mice

When the SAS-3R tumor volume reached 100–150 mm^3^, the mice were randomly categorized into three groups, (a) receiving intravenous (400 μCi/100 μL), (b) intratumoral injection of ^177^Lu-DTPA-pAuNS and (c) intratumoral injection of ^177^Lu-DTPA (200 µCi/50 µL), respectively. After administration of ^177^Lu-DTPA or ^177^Lu-DTPA-pAuNS, we performed microSPECT/CT imaging to acquire signals using the small animal PET/SPECT/CT scanner (FLEX Triumph Regular FLEX X-OCT, SPECT CZT 3 Head System, Gamma Medica, Northridge, CA, USA). Each tumor-bearing mouse was laid in the prone position and anesthetized with 1–2% isoflurane (*w*/*v*) in 2 L oxygen. The images were reconstructed by using an ordered-subsets expectation-maximization (OSEM) algorithm. For imaging conditions, since the scanner has still not yet set the imaging protocol of ^177^Lu, the protocol of ^111^In with similar gamma-ray energy was used for imaging. Each mouse was first positioned by CT scan, then a SPECT scan was performed for 30 min, and the image was reconstructed using OSEM with 20 iterations. We then used Amide’s Medical Image Data Examiner (AMIDE 1.0.4) software to perform semi-automatic region of interest (ROI) drawing and semi-quantitative analysis of the tumor areas in the reconstructed images. After each ROI was circled by AMIDE software, we calculated the cylinder factor obtained from the reference source. The procedure included (a) determining the activity by measuring with a well-type ion chamber, (b) keeping it in a cylindrical container with a fixed volume (cc), (c) scanning the cylinder, reconstructing the image and encircling the ROI and (d) calculating the cylinder factor. The cylinder factor and the injection activity of each mouse were input into AMIDE. Accordingly, we could obtain semi-quantitative data in %ID/cc as the unit.

### 2.6. Evaluation of Therapeutic Efficacy of ^177^Lu-DTPA-pAuNS in Tumor-Bearing Mice

For evaluation of therapeutic efficacy, the tumor viability and growth rate were assessed using the luciferase-based reporter gene imaging (IVIS 50, Perkin Elmer Inc., Waltham, MA, USA) and caliper measurement, respectively. A caliper was used to measure tumor dimensions and its size (mm^3^) was calculated as 0.5 × width^2^ × length. For survival analysis, the endpoint of each datum was established since the first mice died from the treatment. The Kaplan-Meier plot was used for demonstration of survival.

### 2.7. Evaluation of Therapeutic Efficacy of pAuNS-Mediated Optothermal Therapy

The mice in the control and treated group received intratumoral injections of ^177^Lu-DTPA and ^177^Lu-DTPA-pAuNS (10 mg/kg), respectively. After injection, a 6-min laser (793 nm, 1 W/cm^2^) exposure was performed. The increase in temperature was monitored by a digital thermometer using a thermocouple. The treatment response was determined by measuring tumor burden and hematoxylin and eosin staining.

### 2.8. Dosimetric Evaluation of ^177^Lu-DTPA-pAuNS Absorbed Radiation Dose In Vivo

For estimating the radiation absorption dose of various tissues/organs in the human body, the mean absorbed dose in various tissues was calculated from the radionuclide concentration in tissues/organs, and the relative tissue/organ weight was calculated based on previous studies. Based on the results of the biodistribution study in mice, the calculated mean value of the percentage of injected activity per gram of tissue/organ (%IA/g) was extrapolated to present the uptake in organs of a 73 kg adult using the following formula.
[(%IA/g_organ_)_mice_ × (kg_total body weight_)_mice_] × (g_organ_/(Kg_total body weight_)_human_) = (%IA/organ)_human_(1)

The extrapolated values (%IA) in the human organs at 4, 24, 48 and 72 h were exponentially fitted and integrated to obtain the number of disintegrations in source organs. The value of each source organ was entered into the OLINDA/EXM 1.0 software for the dosimetry estimation. For tumor absorbed dose, the number of disintegrations was calculated and entered into the sphere modeling of OLINDA/EXM 1.0 software to estimate the absorbed dose of tumor in each spheroid mass.

### 2.9. Hematoxylin and Eosin (H&E) Staining

Tumor sections were prepared for formalin-fixed paraffin-embedded tissues and selected according to the morphology catalog of the diagnosis. The sections were dewaxed in an oven at 60 °C, and de-paraffinized by xylene, followed by rehydration in an alcohol solution. The tissue sections were stained with hematoxylin and eosin.

### 2.10. Statistical Analysis

The results obtained in this study were analyzed by Student’s *t*-test, and *p* < 0.05 was determined as a significant difference. All data were represented as mean ± standard deviation or mean ± standard error of the mean. The Kaplan-Meier method with the log-rank test was used to compare survival rates among different treatments.

## 3. Results

### 3.1. Design of ^177^Lu Conjugated pAuNS for the Treatment of HNSCC In Vivo

The theranostic radionuclide ^177^Lu was conjugated to the PEGylated AuNS (pAuNS). Figure 1 illustrates the preparation of ^177^Lu conjugated pAuNS via DTPA chelator, the evaluation of tumor accumulation, dosimetry, and therapeutic efficacy after intratumoral administration. The radio TLC analysis showed that the labeling efficiency of ^177^Lu-DTPA-pAuNS was around 87%, and the EDTA challenge with excess EDTA did not significantly affect the labeling efficiency (Figure 1B). After purification by centrifugation, the radiochemical purity (RCP) was 95 ± 3% (Figure 1C). In addition, the radiochemical yield was 50 ± 5%, determined by the radioactivity of the final product over that of initially added ^177^Lu. The TEM visualization demonstrated that the shape and the size of pAuNS were not significantly changed before and after conjugation of ^177^Lu (Figure 1D).

### 3.2. Analysis of ^177^Lu-DTPA-pAuNS Biodistribution in an Orthotopic HNSCC Tumor Model

The biodistribution of intratumorally injected ^177^Lu-DTPA-pAuNS and ^177^Lu-DTPA was quantitatively analyzed. The results showed that the level of ^177^Lu-DTPA-pAuNS was still high in tumor lesions up to 72 h, whereas a little radioactivity was detected in the liver (Figure 2). Free ^177^Lu-DTPA was only detected in tumor lesions for 24 h and also very little in different organs. The tumor-to-muscle (T/M) and tumor-to-blood ratios (T/B) of ^177^Lu-DTPA-pAuNS at different time points were also calculated (Table 1). For intravenous injection of ^177^Lu-DTPA-AuNS, most of the drug was accumulated in the liver and kidney rather than tumors (Appendix A).

### 3.3. MicroSPECT/CT for Evaluation of Intratumoral Injection in HNSCC Tumor

We next compared the tumorous accumulation of ^177^Lu-DTPA and ^177^Lu-DTPA-pAuNS in the orthotopic tumor model. The photon signals of ^177^Lu-DTPA-pAuNS injected tumor were significantly higher than that of ^177^Lu-DTPA injected tumor detected by μSPECT/CT imaging (Figure 3A). No photon signals were detected in other organs of tumor-bearing mice. The imaging result was also semi-quantified by measuring the mean radioactivity in each ROI (Figure 3B). Positive correlations of SPECT imaging and biodistribution were also observed in the tumor uptakes of ^177^Lu-DTPA-pAuNS and ^177^Lu-DTPA (Figure 3C,D).

### 3.4. Evaluation of Therapeutic Efficacy of ^177^Lu-DTPA-pAuNS and ^177^Lu-DTPA in Tumor-Bearing Mice

To evaluate the therapeutic efficacy of ^177^Lu-DTPA-pAuNS in vivo, SAS cells harboring the luciferase reporter gene were used for the establishment of an orthotopic tumor model followed by bioluminescent imaging. The schedule for tumor implantation, drug injection, and in vivo imaging are illustrated in Figure 4A. ^177^Lu-DTPA-pAuNS and ^177^Lu-DTPA (7.4 MBq each) were separately injected into the tumor lesions of small animals. The bioluminescent imaging was acquired using the IVIS 50 system. The results showed that ^177^Lu-DTPA-pAuNS suppressed the luciferase activity, but ^177^Lu-DTPA did not (Figure 4B). The photon flux was quantified to draft the curves for comparison (Figure 4C). The tumor growth curves also demonstrated reduced rates of tumor growth in the ^177^Lu-DTPA-pAuNS treated group (Figure 4D).

### 3.5. Increase of Survival Rate of HNSCC Tumor-Bearing Mice by ^177^Lu-DTPA-pAuNS

The body weights of mice did not decrease more than 80% after treatment of ^177^Lu-DTPA-AuNS or ^177^Lu-DTPA up to 20 days (Figure 5A). Intratumoral injection of ^177^Lu-DTPA-pAuNS showed a higher survival rate in tumor-bearing mice than ^177^Lu-DTPA (Figure 5B). Therefore, ^177^Lu-DTPA-pAuNS has the potential of good therapeutic efficacy by long-term tumor retention and low toxicity using the brachytherapy approach.

### 3.6. Estimation of Dosimetry in Human Organs by the Treatment of ^177^Lu-DTPA-pAuNS

The OLINDA/EXM method was used to estimate the absorbed radiation dose in systemic organs and tumor lesion of a 73 kg adult according to the biodistribution data in HNSCC tumor-bearing mice with intratumoral injection of ^177^Lu-DTPA-pAuNS. The results showed the highest absorbed dose in the liver (0.329 mSv/MBq), kidney (0.135 mSv/MBq), and spleen (0.138 mSv/MBq) (Table 2). The sphere model was used to estimate the absorbed dose of the tumor at 0.5 g (3.55 mGy/MBq), and it is significantly higher than that absorbed by all organs. The ratios of absorbed doses for tumor-to-brain and tumor-to-thyroid were 1517.1 and 2909.8, respectively. The estimated absorbed radiation dose in tumor-bearing mice intravenously injected with ^177^Lu-DTPA-pAuNS (Table 3) was 1.97, 0.16 and 1.37 mSv/MBq, respectively, in the liver, kidney, and spleen. However, the absorbed dose of 0.5 g tumorsphere was only 0.038 mGy/MBq. The ratios of absorbed doses for tumor-to-brain and tumor-to-thyroid were 9.9 and 0.54, respectively. Therefore, the intratumoral injection of ^177^Lu-DTPA-pAuNS was found to be more efficient in the delivery of radiation dose to the tumor but led to less radiation damage to vital organs compared to that by intravenous injection of ^177^Lu-DTPA-pAuNS.

## 4. Discussion

In this study, we applied brachytherapy for the treatment of orthotopic HNSCC tumors using intratumoral injection of ^177^Lu-DTPA-pAuNS. This local chemoradiotherapy method can deliver a relatively precise quantity of high-dose-rate radiopharmaceutical to specific tumor positions so that the risk of side effects is minimized [30,31,32]. Brachytherapy has also been applied in different types of HNSCC, including nasopharyngeal cancer, and oral cancer [33,34]. Although most nanocarriers for in vivo drug delivery are injected through vessels, nanoparticles are notorious for long-term accumulation in the liver and spleen by the reticuloendothelial system [35]. Therefore, the safety of radionuclide-conjugated nanoparticles is inevitably questioned because of radiation-induced damage to vital organs. The brachytherapy approach using intratumoral injection is a highly recommended route to be considered for preclinical and clinical treatment of HNSCC in a tumor model. The biodistribution analysis data have shown only a little dissemination of ^177^Lu-DTPA-pAuNS to normal organs away from the orthotopic tumor by intratumoral injection compared to that by intravenous injection. Because the half-life of ^177^Lu is more than six days, a longer accumulation of ^177^Lu-DTPA-pAuNS in normal organs may possibly endow unexpected side effects [36]. Therefore, intratumoral injection of the nanoparticle-embedded radiopharmaceutical is an interesting approach for HNSCC treatment.

The 5-years survival rate of HNSCC is around 50% after adjuvant chemoradiotherapy [37]. Local invasiveness of HNSCC is associated with EMT and is also a factor that affects the therapeutic efficacy. We have previously shown that intravenous injection of ^188^Re-liposome radiopharmaceutical could induce *let-7i* microRNA in orthotopic HNSCC tumors [29]. Furthermore, RNA-seq analysis revealed that a panel of microRNAs with tumor-suppressive and oncogenic characteristics were up-regulated and down-regulated, respectively, in ^188^Re-liposome treated HNSCC tumors [38]. The expression of EMT molecules was also influenced by ^188^Re-liposome to account for the potential therapeutic efficacy of this radiopharmaceutical [7]. Current data has shown that the use of intratumoral injection led to significant retention of ^177^Lu-DTPA-pAuNS in the HNSCC tumor model. However, little is known about the tumor-suppressive mechanisms of ^177^Lu-DTPA-pAuNS. For precise or personal therapy, it is critical to investigate the therapeutic mechanisms of this novel radiopharmaceutical.

For intravenous injection of ^177^Lu-DTPA-pAuNS, the radioactivity detected in the SAS xenograft tumor was approximately 1.17 ± 0.13%ID/g and 0.59 ± 0.36%ID/g after 24 h and 48 h, respectively (Appendix A). Most of the radioactivity was detected in the liver and spleen. This result is similar to that observed in our previous study, wherein, we used SKOV-3 xenograft tumors injected with ^111^In-DTPA-pAuNS intravenously [26]. Regarding the significant anti-tumoral effect of intravenously delivered AuNS-mediated PTT [26], we expected that the tumor-suppressing effect of intratumorally injected AuNS plus laser exposure should be more apparent because of the higher amount of AuNS accumulated in tumor lesions. Unexpectedly, there were no mice with complete tumor regression based on caliper-measured tumor volumes (Appendix A). However, the H&E staining confirmed that the residual masses were nonviable scar tissues rather than live tumor cells (Appendix A). These results suggest the potential of combining ^177^Lu-DTPA-pAuNS brachytherapy and PTT against HNSCC.

## 5. Conclusions

Current data demonstrate the long retention and therapeutic efficacy of the intratumoral injection of ^177^Lu-DTPA-pAuNS into an orthotopic HNSCC tumor model with little radiation exposure to normal organs. Because AuNS has been applied in PTT for superficial tumor lesions, conjugation of ^177^Lu with AuNS may provide a novel adjuvant brachytherapy approach for cancer treatment.

## Figures and Tables

**Figure 1 pharmaceutics-13-01903-f001:**
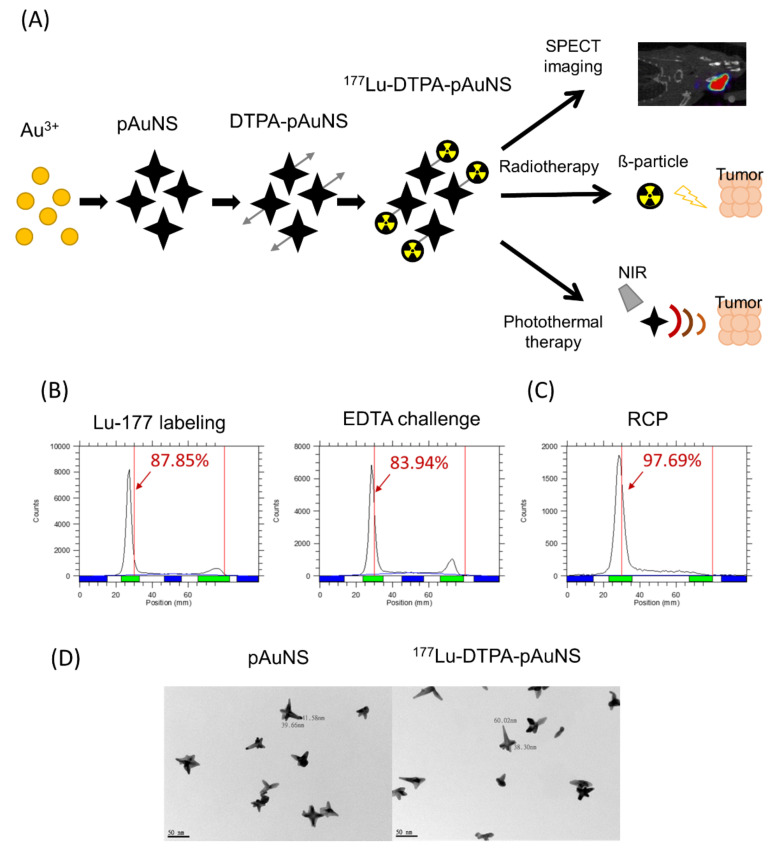
Construction and Radiolabeling of ^177^Lu labeled PEGylated gold nanostars (^177^Lu-DTPA-pAuNS). (**A**) Flow chart of the preparation of ^177^Lu-DTPA-pAuNS and the application of theranostic strategy. (**B**,**C**) RadioTLC analysis shows the labeling efficiency after Lu-177 labeling, EDTA challenge and centrifugation. (**D**) Transmission electron microscopy (TEM) images of pAuNS and ^177^Lu-DTPA-pAuNS.

**Figure 2 pharmaceutics-13-01903-f002:**
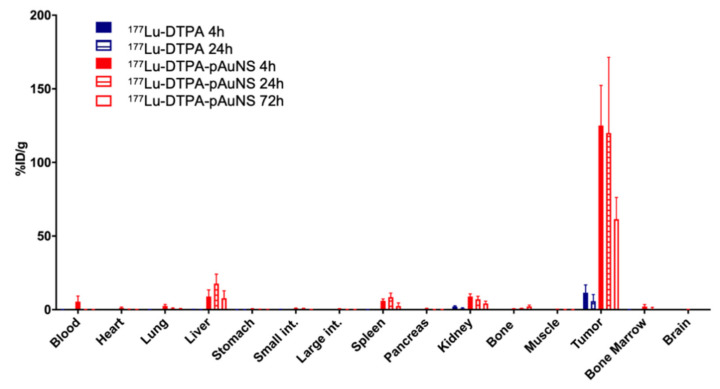
Biodistribution analysis of ^177^Lu-DTPA (Blue bars) and ^177^Lu-DTPA-pAuNS (Red bars) after intratumoral injection in orthotopic HNSCC tumor-bearing mice. Radioactivity distribution in each organ is demonstrated by bar chart (n = 4).

**Figure 3 pharmaceutics-13-01903-f003:**
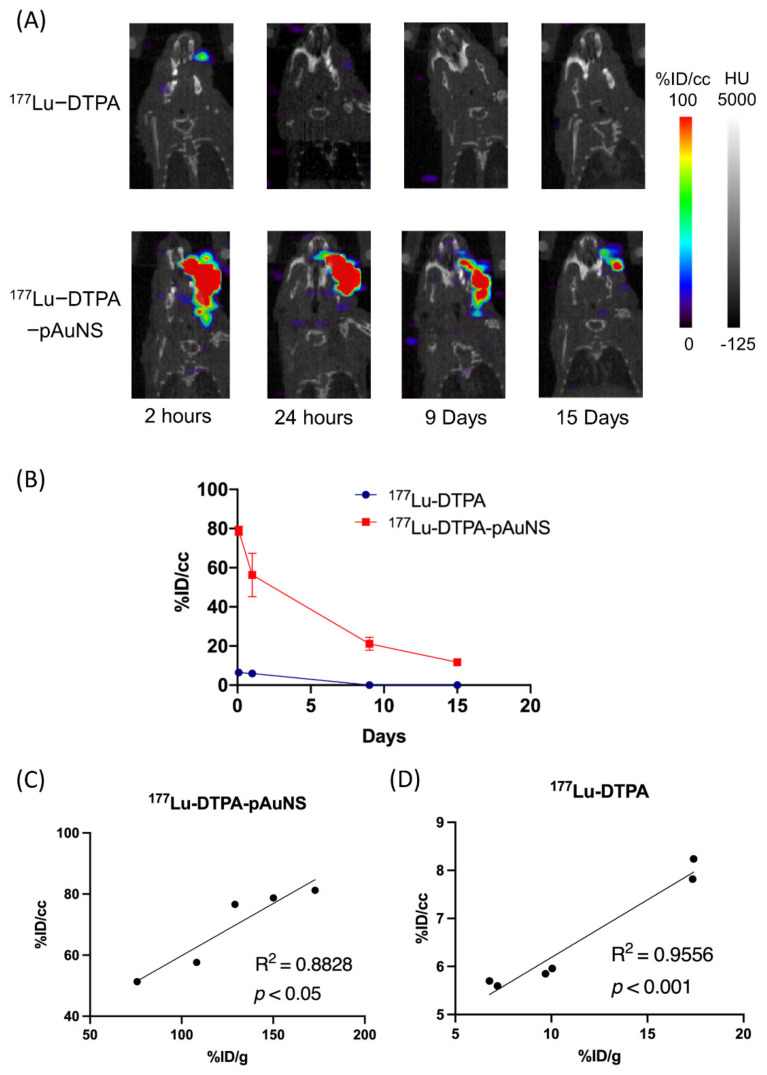
The evaluation of microSPECT/CT in HNSCC orthotopic tumor after intratumoral injection. (**A**) microSPECT/CT imaging of ^177^Lu-DTPA and ^177^Lu-DTPA-pAuNS accumulations in orthotopic HNSCC tumor-bearing mice. (**B**) The time activity curve shows the activity of the tumor area circled by the ROI (n = 3). (**C**,**D**) The correlation of biodistribution analysis (%ID/g) and semi-quantitative analysis (%ID/cc) in ^177^Lu-DTPA-pAuNS and ^177^Lu-DTPA.

**Figure 4 pharmaceutics-13-01903-f004:**
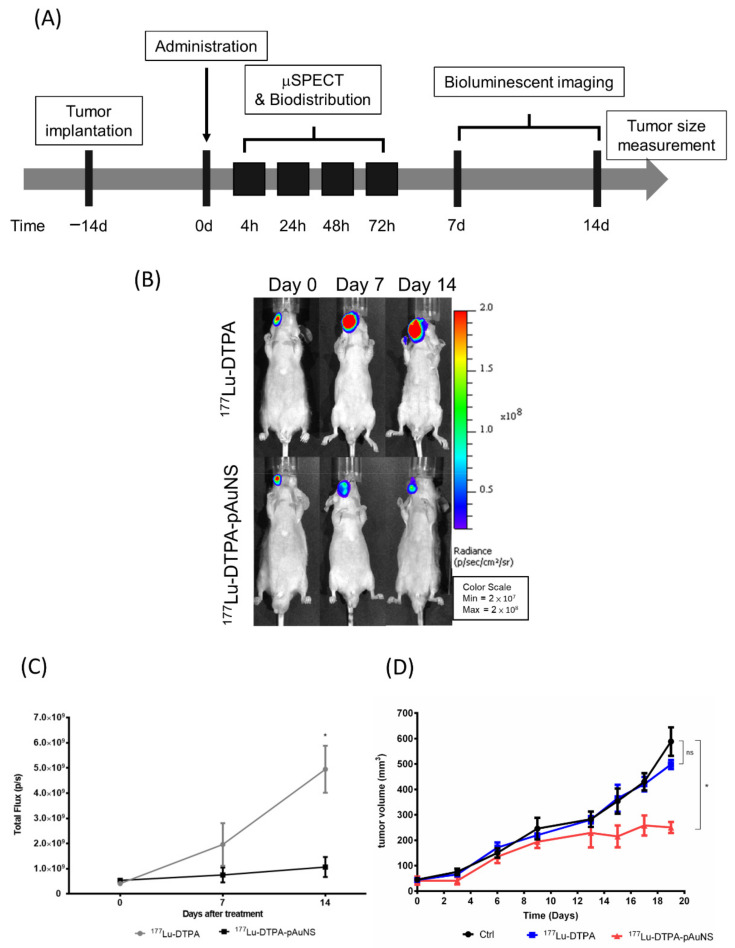
Monitoring the therapeutic efficacy of ^177^Lu-DTPA-pAuNS administrated in HNSCC tumor-bearing mice. (**A**) The experimental scheme for ^177^Lu-DTPA-pAuNS treatment. (**B**) The bioluminescence imaging of tumor growth responding to ^177^Lu-DTPA-pAuNS and untreated control. (**C**) Quantification of the bioluminescence intensity change; *: *p* < 0.05 (**D**) The tumor volume of mice in each group. The mice in the control group, ^177^Lu-DTPA-injected group, and ^177^Lu-DTPA-pAuNSs-injected group were treated with normal saline, 200 μCi of ^177^Lu-DTPA and 200 μCi of ^177^Lu-DTPA-pAuNSs, respectively (n = 5); *: *p* < 0.05; ns: non-significant.

**Figure 5 pharmaceutics-13-01903-f005:**
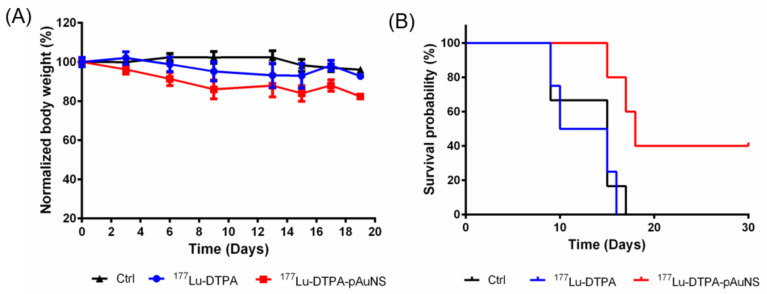
(**A**) The body weight of the mice in each group during the treatment course. The mice in the control group, ^177^Lu-DTPA-injected group, and ^177^Lu-DTPA-pAuNSs-injected group were treated with normal saline, 200 μCi of ^177^Lu-DTPA, and 200 μCi of ^177^Lu-DTPA-pAuNSs, respectively (n = 5). (**B**) The percentage of survival for each group at indicated time points.

**Table 1 pharmaceutics-13-01903-t001:** The biodistribution of ^177^Lu-DTPA-pAuNS injected intratumorally into tumor-bearing mice ^a^.

Organ ^b^	4 h	24 h	48 h	72 h
Blood	5.43 ± 3.78	0.16 ± 0.03	0.21 ± 0.07	0.15 ± 0.02
Heart	1.14 ± 0.65	0.27 ± 0.13	0.23 ± 0.05	0.17 ± 0.09
Lung	2.59 ± 0.95	0.81 ± 0.44	0.62 ± 0.33	0.54 ± 0.27
Liver	8.91 ± 4.50	17.68 ± 6.47	10.43 ± 4.01	7.85 ± 5.02
Stomach	0.59 ± 0.10	0.36 ± 0.07	0.39 ± 0.10	0.31 ± 0.02
Small int.	0.81 ± 0.25	0.61 ± 0.29	0.74 ± 0.38	0.24 ± 0.03
Large int.	0.51 ± 0.17	0.20 ± 0.06	0.25 ± 0.09	0.15 ± 0.07
Spleen	5.99 ± 1.34	8.49 ± 2.74	4.85 ± 2.77	2.53 ± 2.11
Pancreas	0.65 ± 0.25	0.12 ± 0.05	0.13 ± 0.04	0.14 ± 0.03
Kidney	8.97 ± 1.79	7.04 ± 2.07	4.31 ± 0.79	4.23 ± 1.56
Bone	0.65 ± 0.09	0.88 ± 0.06	1.74 ± 0.41	2.20 ± 0.89
Muscle	0.28 ± 0.14	0.07 ± 0.10	0.10 ± 0.05	0.04 ± 0.05
Tumor	125.09 ± 27.26	120.08 ± 51.32	35.62 ± 6.85	61.44 ± 14.81
BM	2.12 ± 1.35	0.55 ± 1.01	2.15 ± 0.86	0.00 ± 0.00
Brain	0.16 ± 0.08	0.02 ± 0.03	0.02 ± 0.01	0.03 ± 0.04
Urine	183.23 ± 81.08	11.41 ± 10.71	2.04 ± 1.10	2.49 ± 2.87
T/M	377.17 ± 154.53	1667.36 ± 1827.93	354.16 ± 152.19	1499.10 ± 279.39
T/B	23.02 ± 41.25	747.62 ± 879.15	169.42 ± 71.88	416.33 ± 110.57

^a^ Unit: percent of injection dose/gram of organ or tissue (%ID/g); ^b^ BM: Bone marrow; T/M: tumor-to-muscle; T/B tumor-to-blood ratio.

**Table 2 pharmaceutics-13-01903-t002:** Radiation dose estimates for intratumoral injection of ^177^Lu-DTPA-pAuNS in an adult human ^a^.

Target Organ ^b^	Absorbed Dose (mSv/MBq)
Adrenals	4.92 × 10^−3^
Brain	2.34 × 10^−3^
Breasts	1.50 × 10^−3^
Gallbladder Wall	7.43 × 10^−3^
LLI Wall	5.06 × 10^−3^
Small Intestine	5.62 × 10^−3^
Stomach Wall	7.17 × 10^−3^
ULI Wall	2.52 × 10^−3^
Heart Wall	2.22 × 10^−2^
Kidneys	1.35 × 10^−1^
Liver	3.29 × 10^−1^
Lungs	2.61 × 10^−2^
Muscle	3.53 × 10^−3^
Ovaries	1.48 × 10^−3^
Pancreas	1.18 × 10^−2^
Red Marrow	2.13 × 10^−3^
Skin	1.25 × 10^−3^
Spleen	1.38 × 10^−1^
Testes	9.93 × 10^−4^
Thymus	1.64 × 10^−3^
Thyroid	1.22 × 10^−3^
Urinary Bladder Wall	1.14 × 10^−3^
Uterus	1.38 × 10^−3^
Tumor (0.5 g) ^c^	3.55 × 10^0^
Total Body	1.69 × 10^−2^
Effective Dose	2.51 × 10^−2^

^a^ The radiation dosimetry was converted from the biodistribution of ^177^Lu-pAuNS in 0.025 kg mice to 73 kg male adults. ^b^ LLI: Lower Large Intestine; ULI: Upper Large Intestine. ^c^ The tumor absorbed dose was obtained by the sphere model. The unit is mGy/MBq because no organ weighting factor is available for the sphere model.

**Table 3 pharmaceutics-13-01903-t003:** Radiation dose estimates for intravenous injection of ^177^Lu-DTPA-pAuNS in an adult human ^a^.

Target Organ ^b^	Absorbed Dose (mSv/MBq)
Adrenals	9.06 × 10^−2^
Brain	3.86 × 10^−3^
Breasts	7.09 × 10^−2^
Gallbladder Wall	1.08 × 10^−1^
LLI Wall	7.77 × 10^−2^
Small Intestine	8.76 × 10^−2^
Stomach Wall	8.14 × 10^−2^
ULI Wall	8.02 × 10^−2^
Heart Wall	2.14 × 10^−2^
Kidneys	1.60 × 10^−1^
Liver	1.97 × 10^0^
Lungs	6.37 × 10^−2^
Muscle	1.14 × 10^−2^
Ovaries	7.40 × 10^−2^
Pancreas	5.01 × 10^−2^
Red Marrow	5.71 × 10^−2^
Skin	6.83 × 10^−2^
Spleen	1.37 × 10^0^
Testes	6.95 × 10^−2^
Thymus	7.19 × 10^−2^
Thyroid	7.03 × 10^−2^
Urinary Bladder Wall	7.23 × 10^−2^
Uterus	7.43 × 10^−2^
Tumor (0.5 g) ^c^	3.82 × 10^−2^
Total Body	1.31 × 10^−1^
Effective Dose	1.71 × 10^−1^

^a^ The radiation dosimetry was converted from the biodistribution of ^177^Lu-pAuNS in 0.025 kg mice to 73 kg male adults. ^b^ LLI: Lower Large Intestine; ULI: Upper Large Intestine. ^c^ The tumor absorbed dose was obtained by the sphere model. The unit is mGy/MBq because no organ weighting factor is available for the sphere model.

## Data Availability

Data generated or analyzed during the study are available from the corresponding author by request.

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
