# Peer review of "Brachytherapy Approach Using 177Lu Conjugated Gold Nanostars and Evaluation of Biodistribution, Tumor Retention, Dosimetry and Therapeutic Efficacy in Head and Neck Tumor Model"

_pharmaceutics, 2021, doi:10.3390/pharmaceutics13111903_

Round 1

Reviewer 1 Report

The study by M.-Y. Lin et al. reports a new radionuclide conjugate for antitumor treatment. The authors designed and synthesized gold nanoparticles with PEG and the radiactive isotope and demonstrated its high efficacy in an orthotopic mouse model of the head and neck tumor. The study is properly designed, the experiments are well performed. Clearly, the new conjugate can be clinically perspective. 

One recommendation before the study can be recommended for publication: I think the text will win if the authors describe the orthotopic model in detail. Please show a survival curve or a tumor volume curve for untreated mice: what is the life span of tumor bearing animals? How do they cope with the growing tumor in terms of nutrition and breath? 

Also, figure legends for Supplementary figures, especially S2, should be described in more detail. What is shown in H&E images? Results of photothermal therapy deserve careful analysis.

What do 'multiple reporters' mean in the description of SAS-3R cell line, and which previous report is mentioned in section 4.2? 

Author Response

Responses to the Reviewer’s Comments

We thank the reviewers for all the suggestions which are helpful for improving this manuscript. We have addressed each comment carefully, and where appropriate, included the requested information in the revised manuscript. The changes in the manuscript have been marked up using the “Track Changes” function. We hope the reviewers and editors will find our response in combination with revision to be satisfactory. The itemized response is as follows:

Reviewers' Comments

Reviewer #1:

The study by M.-Y. Lin et al. reports a new radionuclide conjugate for antitumor treatment. The authors designed and synthesized gold nanoparticles with PEG and the radiactive isotope and demonstrated its high efficacy in an orthotopic mouse model of the head and neck tumor. The study is properly designed, the experiments are well performed. Clearly, the new conjugate can be clinically perspective.

Response: Many thanks to reviewer’s positive comments on this manuscript. We are carefully answering the questions addressed below.

  1. One recommendation before the study can be recommended for publication: I think the text will win if the authors describe the orthotopic model in detail. Please show a survival curve or a tumor volume curve for untreated mice: what is the life span of tumor bearing animals? How do they cope with the growing tumor in terms of nutrition and breath?

Response: We thank the reviewer for highlighting this issue. We have added the information of the tumor volume, body weight, and survival of control (untreated) mice throughout the experiment in Figures 4D, 5A, and 5B, respectively. Generally, we did not detect significant differences in tumor growth rate, body weight loss, and survivals between untreated control and sham control (injected with 177Lu-DTPA without conjugation with AuNS) groups.

Indeed, the growing orthotopic tumors would affect their appetite, especially large tumors born in small animals. Thus, the tumor-bearing mice were fed with food gels instead of food pellets. As a result, severe weakness and body weight loss were minimized in either control or treated groups. Besides, the breathing difficulty was not observed in the orthotopic tumor-bearing mice.

  1. Also, figure legends for Supplementary figures, especially S2, should be described in more detail. What is shown in H&E images? Results of photothermal therapy deserve careful analysis.

Response: We thank reviewer’s valuable comments. We have modified the figure legend of Figure S2 as follows: “Intratumorally delivered pAuNS-mediated photothermal therapy. (A) Thermal imaging of pAuNS-mediated photothermal therapy (PTT) showing the surface temperature of the tumor during laser irradiation. (B) The tumor volume curves of the mice received pAuNS-mediated PTT and controls. (C) H&E staining of residual masses receiving laser alone (left column) and pAuNS-mediated PTT (right column). The nuclei and the cytoplasm were stained with hematoxylin (blue) and eosin (red). T: tumor ; N: non-tumoral surrounding tissues.”

We also added the discussion of findings in PTT in the revised manuscript as follows: “Regarding the significant anti-tumoral effect of intravenously delivered AuNS-mediated PTT [26], we expected that the tumor-suppressing effect of intratumorally injected AuNS plus laser exposure should be more apparent because of the higher amount of AuNS accumulated in tumor lesion. Unexpectedly, there are no mice with complete tumor regression based on capliper-measured tumor volumes (Figure S2A and S2B). However, the H&E staining confirmed that the residual masses were nonviable scar tissues rather than alive tumor cells (Figure S2C). These results suggest the potential of combining 177Lu-DTPA-pAuNS brachytherapy and PTT against HNSCC.” (line 248-256)

Ref: Chen, C.C.; Chang, D.Y.; Li, J.J.; Chan, H.W.; Chen, J.T.; Chang, C.H.; Liu, R.S.; Chang, C.A.; Chen, C.L.; Wang, H.E. Investigation of biodistribution and tissue penetration of PEGylated gold nanostars and their application for photothermal cancer treatment in tumor-bearing mice. J Mater Chem B 2020, 8, 65-77, doi:10.1039/c9tb02194a

  1. What do 'multiple reporters' mean in the description of SAS-3R cell line, and which previous report is mentioned in section 4.2?

Response: We thank the reviewer for pointing out this that we did not well address in the manuscript. The description of SAS-3R cells has been added to the MATERIALS and METHODS section as follows: “Human tongue squamous cell carcinoma SAS cells were obtained from Prof. Muh-Hwa Yang (National Yang Ming Chiao Tung University, Taipei, Taiwan) and SAS-3R cells (a stable cell line with multiple reporter genes, including green fluorescent protein gene, firefly luciferase gene and herpes simplex virus type 1-thymidine kinase gene) harboring a pLT-3R construct with multiple reporter genes were maintained as described in a previous report [37].” (line 278-283)

Ref: Lin, L.T.; Chang, C.Y.; Chang, C.H.; Wang, H.E.; Chiou, S.H.; Liu, R.S.; Lee, T.W.; Lee, Y.J. Involvement of let-7 microRNA for the therapeutic effects of Rhenium-188-embedded liposomal nanoparticles on orthotopic human head and neck cancer model. Oncotarget 2016, 7, 65782-65796, doi:10.18632/oncotarget.11666.

Reviewer 2 Report

In this manuscript, 177Lu-DTPA-pAuNS was prepared and employed as radiopharmaceutical agent for HNSCC brachytherapy. Before acceptance, several issues should be addressed.

  1. The Au Nanostar was used as a carrier to load 177Lu due to its biocompatible and other advantages. However, Au nanorods, Au nanoparticles, iron nanoplatforms, silicon nanoplatforms, etc. also possess the similar properties, why do not employ them as the carrier. Besides, server reports can directly make 177Lu doped in NPs, like ACS Appl. Nano Mater. 2020, 3, 9, 8691–8701. Thus, compared to these NPs, what advantages of your NPs?
  1. The photothermal properties of pAuNS were measured, what about 177Lu-DTPA-pAuNS? Additionally, the tumor after PTT or brachytherapy alone would continue to grow, thus, why do not combine these two therapies? Maybe the synergism of combinational therapy will achieve further enhanced antitumor efficacy.
  1. How about the biosecurity of 177Lu-DTPA-pAuNS after intratumor and intravenously injection?
  2. The author plan to explore the mechanism of this therapy, the following ref. may be meaningful. CHINESE CHEMICAL LETTERS 2021, 32 (5) , pp.1615-1625
  3. To further improve the quality of manuscript, the following references should be cited or discussed. CHINESE CHEMICAL LETTERS, 2021, 32 (1) , pp.158-161. CHINESE CHEMICAL LETTERS, 2021, 32 (3) , pp.1197-1201. CHINESE CHEMICAL LETTERS, 2020, 31 (1) , pp.269-274

Author Response

Responses to the Reviewer’s Comments

We thank the reviewers for all the suggestions which are helpful for improving this manuscript. We have addressed each comment carefully, and where appropriate, included the requested information in the revised manuscript. The changes in the manuscript have been marked up using the “Track Changes” function. We hope the reviewers and editors will find our response in combination with revision to be satisfactory. The itemized response is as follows:

Reviewers' Comments

Reviewer #2:

In this manuscript, 177Lu-DTPA-pAuNS was prepared and employed as radiopharmaceutical agent for HNSCC brachytherapy. Before acceptance, several issues should be addressed.

  1. The Au Nanostar was used as a carrier to load 177Lu due to its biocompatible and other advantages. However, Au nanorods, Au nanoparticles, iron nanoplatforms, silicon nanoplatforms, etc. also possess the similar properties, why do not employ them as the carrier. Besides, server reports can directly make 177Lu doped in NPs, like ACS Appl. Nano Mater. 2020, 3, 9, 8691–8701. Thus, compared to these NPs, what advantages of your NPs?

Response: We thank the reviewer for highlighting this issue. Indeed, there are several nanoparticles with superior accumulation in tumor lesions, such as gold nanorods, iron nanoplatforms, silicon nanoparticles, and so on. The main reason for using gold nanostars in the present study is that the gold nanostars possess a significant photothermal conversion rate for photothermal therapy compared to other nanoparticles [26]. The scope of this study is to determine the characteristics of 177Lu-labeled gold nanostars. We expect that 177Lu-DTPA-pAuNSs would be used as a therapeutic agent for photothermal-brachytherapy in tumor treatment. In this proof-of-concept study, the potential of 177Lu-DTPA-pAuNSs, either in brachytherapy or PTT, has been shown in Figures 4, 5, and S2. We are going to launch the experiments to comprehensively determine the therapeutic efficacy of 177Lu-pAuNS-mediated PTT in combination with brachytherapy in the future.

Ref: Chen, C.C.; Chang, D.Y.; Li, J.J.; Chan, H.W.; Chen, J.T.; Chang, C.H.; Liu, R.S.; Chang, C.A.; Chen, C.L.; Wang, H.E. Investigation of biodistribution and tissue penetration of PEGylated gold nanostars and their application for photothermal cancer treatment in tumor-bearing mice. J Mater Chem B 2020, 8, 65-77, doi:10.1039/c9tb02194a

  1. The photothermal properties of pAuNS were measured, what about 177Lu-DTPA-pAuNS? Additionally, the tumor after PTT or brachytherapy alone would continue to grow, thus, why do not combine these two therapies? Maybe the synergism of combinational therapy will achieve further enhanced antitumor efficacy. How about the biosecurity of 177Lu-DTPA-pAuNS after intratumor and intravenously injection?

Response: We thank the reviewer’s valuable comments. Based on our experience, the main factors for successful photothermal conversion are the size and shape of nanoparticles. Because no significant difference in size and shape between pAuNSs and 177Lu-DTPA-pAuNSs (Figure 1D), the photothermal conversion ability of 177Lu-DTPA-pAuNSs would be similar to that of pAuNSs.  

We agree that 177Lu-pAuNS-mediated PTT plus brachytherapy should achieve better tumor control. In this manuscript, however, we focused on the anti-tumor ability of 177Lu-DTPA-pAuNS mediated brachytherapy because the therapeutic property of this radiopharmaceutical remains unclear. Although the combined PTT therapy is beyond the scope of this study, the related study is important and is going to be evaluated in the near future.

For the issue of biosafety of 177Lu-DTPA-pAuNSs, based on the dosimetry analysis, the effective dose for intratumoral injection and intravenous injection of 177Lu-DTPA-pAuNSs were 2.51 × 10-2 and 1.71 × 10-1 mSv/MBq, respectively. If we intratumoral deliver 100 MBq of 177Lu-DTPA-pAuNSs, the resulting dose is just similar to the annual dose (~ 2.5 mSv in Taiwan) from natural background radiation, which may not cause any safety issue. Besides, we did not notice severe weakness and body weight loss in each group (Figure 5A), suggesting the 177Lu-induced radiation burden is not the critical issue in the brachytherapy of 177Lu-DTPA-pAuNSs.      

  1. The author plan to explore the mechanism of this therapy, the following ref. may be meaningful. CHINESE CHEMICAL LETTERS 2021, 32 (5) , pp.1615-1625. To further improve the quality of manuscript, the following references should be cited or discussed. CHINESE CHEMICAL LETTERS, 2021, 32 (1) , pp.158-161. CHINESE CHEMICAL LETTERS, 2021, 32 (3) , pp.1197-1201. CHINESE CHEMICAL LETTERS, 2020, 31 (1) , pp.269-274?

Response: We thank the reviewer for the valuable comments. We have read your suggested relevant papers and cited them all in the DISCUSSION section of the revised manuscript (Ref. 24, 30, 31, and 35).

Reviewer 3 Report

The Authors performed a pivotal study synthetizing a novel radiopharmaceutical by conjugating the b − -emitter 177Lu with  DTPA-polyethylene glycol (PEG) decorated gold nanostar (177Lu-DTPA-pAuNS). They studied accumulation, therapeutic efficacy and compared these characteristics with those of 177Lu-DTPA on an 25 orthotopic tumor model of head&neck squamous cell carcinoma. This a timely and well conducted study. Few comments:

  • Line 28. Correct typo: injection.
  • Line 35. Maybe add potential (novel radiophamaceutical).
  • Line 39. Not that ‘common’. Maybe add some numerical data
  • Line 40. What do you mean by ‘moderate’ for 5-year OS? Please use numbers.
  • Line 42. Torso? Please change.
  • Line 66,67,68 please correct typo.
  • Line 87-96. Please provide a rationale for the study, but try to shorten the paragraph, without providing details which will be pointed out in the subsequent paragraphs.
  • Discussion: Line 218-219. I would not define eye and oesophageal cancers as HN cancer. Please change.

Author Response

Responses to the Reviewer’s Comments

We thank the reviewers for all the suggestions which are helpful for improving this manuscript. We have addressed each comment carefully, and where appropriate, included the requested information in the revised manuscript. The changes in the manuscript have been marked up using the “Track Changes” function. We hope the reviewers and editors will find our response in combination with revision to be satisfactory. The itemized response is as follows:

Reviewers' Comments

Reviewer #3:

The Authors performed a pivotal study synthetizing a novel radiopharmaceutical by conjugating the b − -emitter 177Lu with  DTPA-polyethylene glycol (PEG) decorated gold nanostar (177Lu-DTPA-pAuNS). They studied accumulation, therapeutic efficacy and compared these characteristics with those of 177Lu-DTPA on an 25 orthotopic tumor model of head&neck squamous cell carcinoma. This a timely and well conducted study. Few comments:

Response: Many thanks to the reviewer’s positive comments on this manuscript. We are carefully answering the questions point-by-point:

  1. Line 28. Correct typo: injection.

Response: We thank the reviewer for the kind reminder. We have corrected it (line 28).

  1. Line 35. Maybe add potential (novel radiophamaceutical).

Response: We thank the reviewer for the valuable comments. We have revised the description as your suggestion (line 35).     

  1. Line 39. Not that ‘common’. Maybe add some numerical data.

Response: We thank the reviewer for the valuable comments. We have re-written the sentence as follows: “Head and neck squamous cell carcinoma (HNSCC) ranks sixth in incidence around the world.” (line 39)

  1. Line 40. What do you mean by ‘moderate’ for 5-year OS? Please use numbers.

Response: We thank the reviewer for the valuable comments. We have re-written the sentence as follows: “The 5-year survival rate of HNSCC is around 65% and ….” (line 40)   

  1. Line 42. Torso? Please change.

Response: We thank the reviewer for the valuable comments. We have revised this sentence as follows: “Unlike other cancers occurred in parenchymal organs, the dissemination of HNSCC mainly caused by local invasion instead of distant metastasis.” (lines 42-43)

  1. Line 66,67,68 please correct typo.

Response: We thank the reviewer for the valuable comments. We have corrected them. (lines 66-68)

  1. Line 87-96. Please provide a rationale for the study, but try to shorten the paragraph, without providing details which will be pointed out in the subsequent paragraphs.

Response: We have re-written this paragraph in the INTRODUCTION section as follows: “Based on the promising theranostic properties of 177Lu and the nature of the gold nanostars, we aim to develop PEGylated coated AuNS for conjugating 177Lu via a DTPA chelator and determine their biodistribution using noninvasive imaging as well as the therapeutic efficacy. To the best of our knowledge, this is the first report proposing the preclinical application of 177Lu labeled AuNS in the human HNSCC tumor model.” (lines 87-91).

  1. Discussion: Line 218-219. I would not define eye and oesophageal cancers as HN cancer. Please change

Response: We thank the reviewer for the valuable comments. We have deleted “eye cancer” and “esophageal cancer” from this sentence (lines 216-217).
